# Deep Graph Pose: a semi-supervised deep graphical model for improved animal pose tracking

**Anqi Wu**[1][*]    **E. Kelly Buchanan**[1][*]    **Matthew R Whiteway**[1]    **Michael Schartner**[2]

**Guido Meijer**[3]    **Jean-Paul Noel**[4]    **Erica Rodriguez**[1]    **Claire Everett**[1]

**Amy Norovich**[1]    **Evan Schaffer**[1]    **Neeli Mishra**[1]    **C. Daniel Salzman**[1]

**Dora Angelaki**[4]    **Andrés Bendesky**[1]    **The International Brain Laboratory**[5]

**John Cunningham**[1]    **Liam Paninski**[1]

[1] Columbia University, New York, USA
{aw3236, ekb2154, mw3323, er2934, cpe2108, aln2128, ess2129, nm2786
cds2005, ab4463, jpc2181, lmp2107}@columbia.edu
[2] University of Geneva, Geneva, Switzerland
Michael.Schartner@unige.ch
[3] The Champalimaud Centre for the Unknown, Lisbon, Portugal
guido.meijer@research.fchampalimaud.org
[4] New York University, New York, USA
{jpn5, da93}@nyu.edu
[5] info@internationalbrainlab.org

## Abstract

Noninvasive behavioral tracking of animals is crucial for many scientific investigations. Recent transfer learning approaches for behavioral tracking have considerably advanced the state of the art. Typically these methods treat each video frame and each object to be tracked independently. In this work, we improve on these methods (particularly in the regime of few training labels) by leveraging the rich spatiotemporal structures pervasive in behavioral video — specifically, the spatial statistics imposed by physical constraints (e.g., paw to elbow distance), and the temporal statistics imposed by smoothness from frame to frame. We propose a probabilistic graphical model built on top of deep neural networks, Deep Graph Pose (DGP), to leverage these useful spatial and temporal constraints, and develop an efficient structured variational approach to perform inference in this model. The resulting semi-supervised model exploits both labeled and unlabeled frames to achieve significantly more accurate and robust tracking while requiring users to label fewer training frames. In turn, these tracking improvements enhance performance on downstream applications, including robust unsupervised segmentation of behavioral "syllables," and estimation of interpretable "disentangled" low-dimensional representations of the full behavioral video. Open source code is available at https://github.com/paninski-lab/deepgraphpose.

---

[*]equal contribution

# 1 Introduction

Animal pose estimation (APE) is a critical scientific task, with applications in ethology, psychology, neuroscience, and other fields. Recent work in neuroscience, for example, has emphasized the degree to which neural activity throughout the brain is correlated with movement [1, 2, 3]; i.e., to understand the brains of behaving animals we need to extract as much information as possible from behavioral video recordings. State of the art APE methods, such as DeepLabCut (DLC) [4], DeepPoseKit (DPK) [5], and LEAP [6], have transferred tools from human pose estimation (HPE) in deep learning literature to the APE setting [7, 8], opening up an exciting array of new applications and new scientific questions to be addressed.

However, even with these advances in place, hundreds of labels may still be needed to achieve tracking at the desired level of precision and reliability. Providing these labels requires significant user effort, particularly in the common case that users want to track multiple objects per frame (e.g., all the fingers on a hand or paw). Unlike HPE algorithms [9], APE algorithms are applied to a wide variety of different body structures (e.g., fish, flies, mice, or cheetahs) [10], compounding the effort required to collect labeled datasets and hindering our ability to re-use a common skeletal model. Moreover, even with hundreds of labels, users still often see occasional "glitches" in the output (i.e., frames where tracking is briefly lost), which typically interfere with downstream analyses of the extracted behavior.

To improve APE performance in the sparse-labeled-data regime, we propose a probabilistic graphical model built on top of deep neural networks, Deep Graph Pose (DGP), to leverage both spatial and temporal constraints, and develop an efficient structured variational approach to perform inference in this model. DGP is a semi-supervised model that takes advantage of both labeled and unlabeled frames to achieve significantly more accurate and robust tracking, using fewer labels. Finally, we demonstrate that these tracking improvements enhance performance in downstream applications, including robust unsupervised segmentation of behavioral "syllables," and estimation of interpretable low-dimensional representations of the full behavioral video.

# 2 Related Work

**Animal pose estimation**. The proposed approach fills a void between state of the art human pose estimation algorithms, which often rely on large quantities of manually labeled samples (see [9] for a recent review), and their counterparts in animal pose estimation [11, 4, 6, 5, 12, 13]. Among these animal pose estimation algorithms, DLC [4], LEAP [6], and DPK [5] stand out as they can achieve near human-level accuracy. However, all these methods rely on a large number of human labels in order to achieve the desired level of precision and reliability. Our work extends such models with a probabilistic graphical model that use unlabeled frames and temporal and spatial structures. [14] has recently proposed to incorporate temporal context from nearby video frames using optical flow which occurs only at the test stage to refine the model's predictions. However, in our approach, we incorporate the temporal context into the trainable graphical model.

**Graphical models**. Previous work on human pose estimation has employed graphical models as regularizers for convolutional networks [15, 16, 17, 18, 19, 20]. Among these, [17] and [18], like DGP, build an undirected graphical model (UGM) on top of deep neural networks. However, unlike DGP, they assign tracked locations discrete values, which allows for (discrete) message passing algorithms during the inference step. [19] builds a spatial-temporal graph similar to DGP. But none of these previous methods uses unlabeled frames to improve performance, as DGP does. They were all proposed for human pose estimation which has many benchmark datasets with a large number of labels. [20] has proposed a method later for sparsely-labeled videos but without any spatial constraints.

**Semi-supervised learning**. Semi-supervised learning aims to fully utilize unlabeled or weakly-labeled data to gain additional insights into the structure of the data [21, 22, 23]. Many pose estimation algorithms have adopted such learning schemes to enhance the performance given limited training data [24, 25]. One conceptually similar "weakly-supervised" approach is described by [26], who trained a network to extract flying objects (obeying Newtonian acceleration) simply by constraining the output to resemble a parabola. In our work, DGP encourages the output confidence

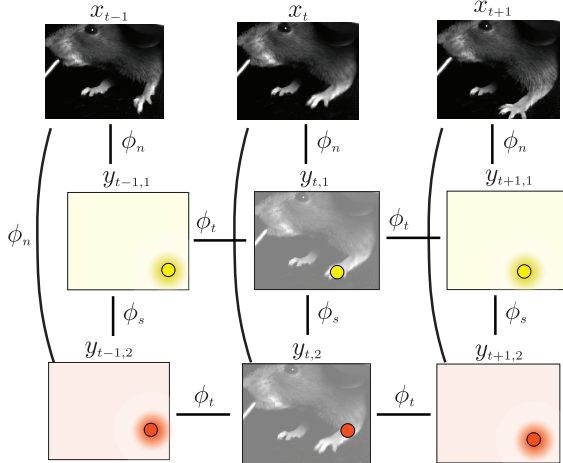

Figure 1: **Deep Graph Pose (DGP) model**. DGP leverages observed (labeled) and hidden information to infer the locations of unobserved targets via graph semi-supervised inference. At each time $t$, we observe the frame $x_t$. We want to track multiple targets in each frame (in this case, the paw and elbow). We also observe the labels of the two targets in some frames (in this example, in the $t$-th frame), denoted as $y_{t,1}$ and $y_{t,2}$ (colored circles at $t$). The hidden variables are the unobserved targets (indicated with colored circles in the colored background in frames $t-1$ and $t+1$ here).

map to be unimodal; this can be seen as a form of weak supervision that leads to improved accuracy even when the temporal and spatial soft constraints are removed.

## 3 Model

The graphical model of DGP is summarized in Figure 1. We observe frames $x_t$ indexed by $t$, along with a small subset of labeled markers $y_{t,j}$ (where $j$ indexes the different targets we would like to track). The target locations $y_{t,j}$ on most frames are unlabeled, but we have several sources of information to constrain these latent variables: temporal smoothness constraints between the targets $y_{t,j}$ and $y_{t+1,j}$ , which we capture with potentials $\phi_t$; spatial constraints between the targets $y_{t,i}$ and $y_{t,j}$, which we model with spatial potentials $\phi_s$; and information from the image $x_t$, modeled by $\phi_n$.

We parametrize $\phi_n$ with a neural network, indicated by the subscript $n$. A number of architectures could potentially be employed for $\phi_n$ [6, 5]; we chose to adapt the architecture used in DLC [4] here.

For simplicity, we start with a quadratic potential $\phi_t$ to impose temporal smoothness:

$$\phi_t^j(y_{t,j}, y_{t+1,j}) \quad = \quad \frac{1}{2} w_t^j ||y_{t,j} - y_{t+1,j}||^2, \tag{1}$$

which penalizes the distance between targets in consecutive frames; the weights $w_t^j$ in general may depend on the target index $j$, and can also vary in time. A quadratic potential is equivalent to modeling the target at the next time step as normally distributed around the current target, which is also equivalent to Gaussian random walk. We will discuss extensions of this simple quadratic potential in the appendix.

The spatial potential $\phi_s$ is more dataset-dependent and can be chosen depending on the constraints that the markers should satisfy. Typical examples include a soft constraint that the paw marker should not exceed some distance from the elbow marker, or the nose should always stay within a certain radius of a static waterspout. Again, we start with a simple quadratic potential to encode these soft constraints:

$$\phi_s^{ij}(y_{t,i}, y_{t,j}) \quad = \quad \frac{1}{2} w_s^{ij} ||y_{t,i} - y_{t,j}||^2, \tag{2}$$

which penalizes the distance between "connected" targets $y_{t,i}$ and $y_{t,j}$ (where the user can pre-specify pairs of connected targets that should have neighboring locations in the frame, e.g. paw and elbow); more sophisticated non-quadratic losses are again discussed in the appendix.

We want to "let the data speak" and avoid oversmoothing, so the penalty weights $w_s$ and $w_t$ should be small. In practice we found that the temporal weights $w_t^j$ could be set using optical flow [27] which captures the vector field between adjacent frames. We first computed the vector field between two neighbor frames $t-1$ and $t$ using optical flow. Then we calculated the average motion vector for target $j$ from frame $t-1$ to frame $t$. The magnitude of the motion vector was denoted as $m_t^j$. Finally $w_t^j = \xi/m_t^j$, where $\xi$ is a constant scalar independent of dataset, time and target indices. The

intuition is the larger the movement of the target is, the smaller the temporal clique weight should be. We set the spatial weights as $w_s^{ij} = c/d_{ij}$, where $d_{ij}$ is a rough estimate of the average distance (in pixels) between targets $i$ and $j$ and $c > 0$ is a small scalar (again independent of dataset and target indices $i, j$), which led to robust results without any need to fit extra parameters. We summarize the parameter vector as $\beta = \{\theta, w_t, w_s\}$, where $\theta$ denotes the neural net parameters in $\phi_n$. Given $\beta$, the joint probability distribution over targets $y$ is

$$
\begin{aligned}
p(y|x,\beta) = \frac{1}{Z(x,\beta)} \exp\bigg( &- \sum_{t=1}^{T} \sum_{j=1}^{J} \phi_n^j(y_{t,j}, x_t) \\
&- \sum_{t=1}^{T-1} \sum_{j=1}^{J} \phi_t^j(y_{t,j}, y_{t+1,j}) - \sum_{t=1}^{T} \sum_{i,j \in \mathcal{E}} \phi_s^{ij}(y_{t,i}, y_{t,j}) \bigg),
\end{aligned}
\tag{3}
$$

where $\mathcal{E}$ denotes the edge set of constrained targets (i.e., the pairs of markers $i, j$ with a nonzero potential function), $Z(x,\beta) = \int p(y|x,\beta)dy$ is the normalizing constant marginalizing out $y$, $T$ denotes the total number of frames, and $J$ denotes the total number of targets.

# 4 Structured variational inference

Our goal is to estimate $p(y^h \mid y^v, x, \beta)$, the posterior over locations of unlabeled targets $y^h$, given the frames from the video $x$, the locations of the labeled markers $y^v$, and the parameters $\beta$. Here $h$ denotes hidden, for the unlabeled data, and $v$ denotes visible, for the labeled data. Calculating this posterior distribution exactly is intractable, due to the highly nonlinear convolutional networks appearing in potentials $\phi_n$. We chose to use structured variational inference [28, 29] to approximate this posterior. We approximate $p(y^h, y^v \mid x, \beta)$ with a Gaussian graphical model (GGM) with the same graphical model as Figure 1, leading to a Gaussian posterior approximation $q(y^h \mid y^v, x, \beta)$ for $p(y^h \mid y^v, x, \beta)$ in which the inverse covariance (precision) matrix is block tridiagonal (Gaussian random walk), with one block per frame $t$. Since the potentials $\phi_t$ and $\phi_s$ are quadratic, yielding Gaussian distributions, the neural-network image potential $\phi_n$ is the only term that needs to be replaced with a new quadratic potential to form a Gaussian $q$.

Updating the parameters of this GGM scales as $O(TJ^3)$ in the worst case, due to the chain structure of the graphical model (and the corresponding block tridiagonal structure of the precision matrix). If the edge graph $\mathcal{E}$ defined by the user-specified spatial potential function set is disconnected, this $J^3$ factor can be replaced by $K^3$, where $K$ is the size of the largest connected component in $\mathcal{E}$.

We used a structured inference network approach [29] to estimate the model and variational parameters. We computed gradients of the evidence lower bound (ELBO) for this model using standard automatic differentiation tools, and performed standard stochastic gradient updates to estimate the parameters. Full details regarding the ELBO derivation and optimization can be found in Section S1 in the appendix.

## 4.1 Conceptual comparison against fully-supervised approaches

Standard fully-supervised approaches like DeepLabCut [4] learn a neural network (or more precisely, use transfer learning to adjust the parameters of an existing neural network) to essentially perform a classification task: the network is trained to output large values at the known location of the markers (i.e., the "positive" training examples), and small values everywhere else (the "negative" training examples). Given a small number of training examples, these methods are prone to overfitting.

In contrast, the approach we propose here is semi-supervised: it takes advantage of both the labeled and unlabeled frames to learn better model parameters $\theta$. On labeled frames, the posterior distribution $p(y^v \mid y^v, x, \beta)$ is deterministic, and the objective function reduces to the fully supervised case. On the other hand, on unlabeled frames we have new terms in the objective function (see section S1.2.1 for more details). Clearly, the spatial and temporal potentials $\phi_s$ and $\phi_t$ encourage the outputs to be temporally smooth and to obey the user-specified spatial constraints (at least on average). But in addition the objective function encourages $\phi_n$ to output large values where $p(y^h \mid y^v, x, \beta)$ is large, and small values where $p(y^h \mid y^v, x, \beta)$ is small. Since we approximate $p(y^h \mid y^v, x, \beta)$ as Gaussian, the resulting ELBO encourages $\phi_n$ to be (on average) unimodal on unlabeled frames — a constraint

Table 1: Dataset summary.

| Dataset | Brief Description | Dimensions (x, y, t) | Number of labeled frames |
|---|---|---|---|
| mouse-wheel [30] | moving a wheel | (374, 450, 1000) | 55 |
| mouse-reach [31] | grabbing a stick | (747, 832, 256) | 52 |
| fly-run [32] | running on a ball | (600, 600, 1210) | 13 |
| twomice-top-down* | freely moving | (480, 640, 1364) | 20 |
| fish-swim [33] | freely swimming | (471, 475, 2000) | 20 |

(*) unpublished

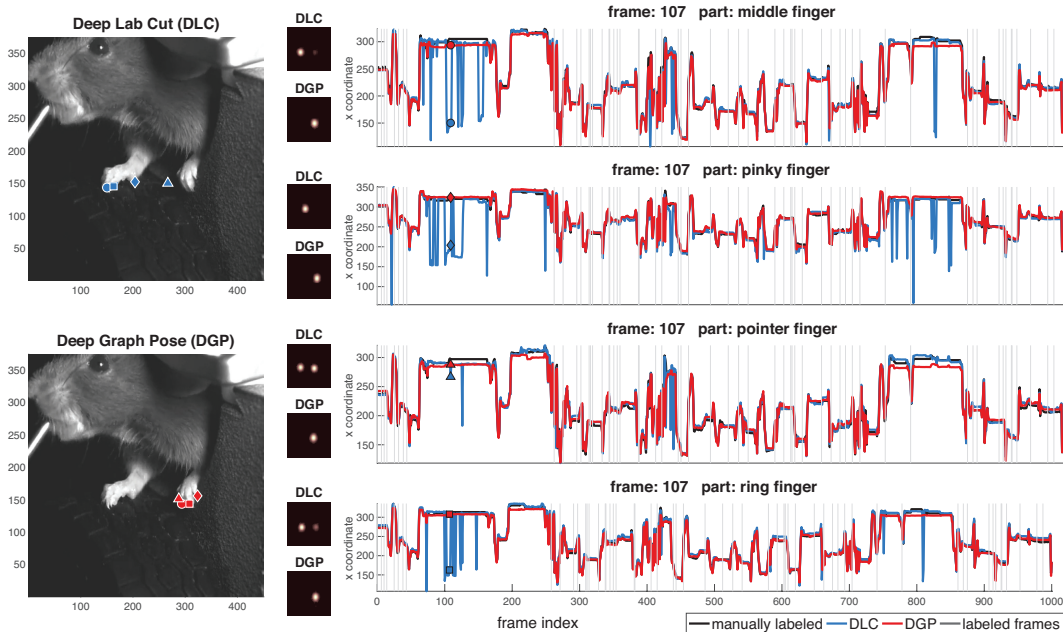

Figure 2: **Comparison of Deep Graph Pose (DGP) versus DeepLabCut (DLC) and manually-labeled data on the mouse-wheel dataset from [30]**; see also [34]. Left panels show an example frame, with the DLC output markers superimposed in blue (top) and the DGP markers in red (bottom). The right panels show the horizontal marker positions as a function of time (with DLC in blue, DGP in red and the full manually-labeled trace in black). Vertical lines indicate labeled (training) frames. The small inset images show confidence maps for each marker output by DLC (top) and DGP (bottom); the DGP confidence maps tend to be more unimodal than the DLC confidence maps. Note that the DLC and DGP marker locations tend to agree on labeled frames, but we see significant discrepancies on unlabeled test frames. Visual inspection of the videos (and comparison again the manual labels) indicates that when the DLC and DGP markers disagree, typically the DLC marker is in the wrong location.

that is not enforced in standard approaches. This turns out to be a powerful regularizer and can lead to significant improvements even in cases where the spatial and temporal constraints $\phi_s$ and $\phi_t$ are weak, as we will see in the next section.

## 5   Results

We applied DGP and DLC[2] to a variety of datasets, including behavioral videos from three different species, in a variety of poses and environments (see Table 1 for a summary). The new model (DGP) consistently outperformed the baseline (DLC). In each example video analyzed here, DLC outputs

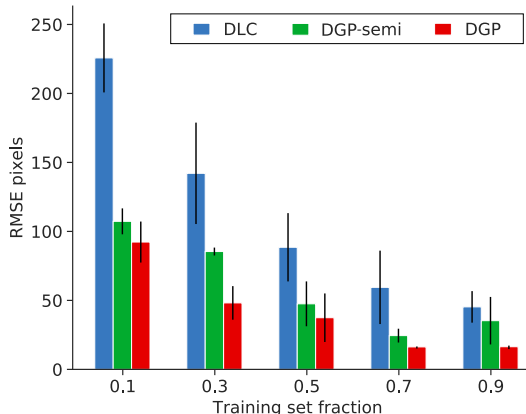

Figure 3: **Quantification of the results from Figure 2 over multiple training set sizes and ablation experiments**. DGP outperforms DLC and the intermediate variant DGP-semi. We evaluated the different methods (see main text for definition of DGP-semi) using multiple random subsets of the training set (55 labels) and compared the differences in test error. Error bars represent one standard error across five random trials. Each random trial has its own randomly generated training set.

occasional "glitch" frames where tracking of at least one target was lost (e.g., around frame index 100 in the lower right panel); these glitches were much less prevalent in the DGP output. We experimented with running Kalman smoothers and total variation denoisers to post-process the DLC output, but were unable to find any parameter settings that could reliably remove these glitches without oversmoothing the data (results not shown). The frequency of these "glitches" can be reduced by increasing the training set through labeling more data — but this is precisely the user effort we aim to minimize here. See the full videos summarizing the performance of the two methods. An example screenshot for the mouse-wheel dataset [30] is shown in Figure 2. The comparison between DLC and DGP on all other datasets can be found in Figures S3-S6 in the appendix. More information regarding experimental setup can be found in Section S4 in the appendix.

We also examined the "confidence maps" generated by visualizing the output of the neural network $\phi_n$ as an image; large values of the confidence map indicated the regions where the network "believed" the target was located with high confidence. Comparing the confidence maps output by DLC versus DGP, we see that the latter tended to be more unimodal (see Figure 2, small panels in the middle column). Nonetheless, DGP did occasionally output multi-modal confidence maps (e.g., in frames where the target was occluded), since the ELBO objective function used to train DGP encouraged unimodality but did not impose unimodality as a hard constraint.

To better understand the source of the performance gains exhibited by DGP, we also experimented with a model in which the spatial and temporal potentials were turned off (i.e., $w_s = w_t = 0$). The resulting graphical model can be factorized over targets $j$ and frames $t$. We call the resulting model DGP-semi, since the resulting ELBO objective function combines a usual supervised loss (as in DLC) with an unsupervised term that encourages the output of the image potential $\phi_n$ to match its Gaussian approximation for each $(t, j)$ pair (i.e., the resulting loss can be considered a semi-supervised hybrid model). Comparing DLC, DGP-semi, and DGP provides a qualitative sense of the relative benefits of the semi-supervised loss and the spatial and temporal cliques (see videos).

To develop more quantitative comparisons, we manually labeled 1000 frames in the mouse-wheel dataset[3]. We randomly assigned 55 labeled frames to the training set and used the remaining 945 frames as the test set. Next we randomly subsampled 10%–90% of this training set and retrained the models to quantify the relation between the test errors and the number of labeled frames. Figure 3 shows the test errors averaged over five random subsamples. We see that DGP-semi and DGP outperformed DLC uniformly over the training set fractions (i.e., the number of labeled frames used to train the model) with a significant amount of improvement. DGP further decreased the errors with the extra spatial and temporal constraints. Similar results were obtained using an $\epsilon$-insensitive loss that ignored errors below a threshold $\epsilon$ (on the order of 5-10 pixels here) below which the "true" marker location becomes somewhat subjective (results not shown here).

From both qualitative and quantitative analyses, we can tell that although DGP-semi does not enforce any spatial constraints or temporal smoothness, the extra regularization from the unsupervised term

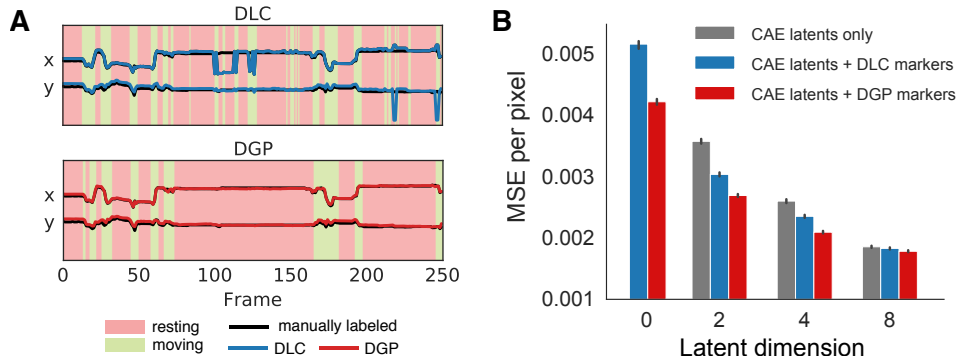

Figure 4: (A) **Unsupervised methods segment DGP traces into interpretable "resting" versus "moving" states, while DLC trace segmentation is hampered by glitches**. We ran a two-state autoregressive hidden Markov model (ARHMM) on the DGP and DLC outputs (in this case, on the x- and y-coordinates of a single paw). Background colors indicate the inferred states from the ARHMM fit to the DGP or DLC traces. The model fit with the DGP output clearly learns interpretable states, a "resting" state (red) and a "moving" state (green) (bottom). The model fit with the DLC output learns two states that are partially corrupted by "glitches" where DLC jumps away from the manually-labeled paw position (bottom); see video for full details. (B) **Conditioning CAEs on DGP markers improves reconstruction performance.** We computed mean square error (MSE) per pixel on reconstructed test frames from the mouse-wheel dataset when using a CAE (gray bars), or conditional CAEs, where the markers output by DLC (blue) or DGP (red) are used as input to both the encoder and decoder networks. A latent dimension of 0 corresponds to directly decoding the frames from markers. We see that test MSE decreases with latent dimensionality (as expected), and that the model conditioned on DGP markers consistently outperforms the model conditioned on DLC markers. Error bars represent 95% bootstrapped confidence interval over test frames. Reconstruction videos are also available.

in the ELBO encourages the model output to be more unimodal, leading to significantly improved predictions compared to DLC. With the additional temporal and spatial constraints, DGP can further improve the performance.

## 5.1 Downstream analyses

The above results demonstrate that DGP provides improved tracking performance compared to DLC. Next we show that these accuracy improvements can in turn lead to more robust and interpretable results from downstream analyses based on the tracked output.

**Unsupervised temporal segmentation**. We begin with a segmentation task: given the estimated trace for the paw, can we use unsupervised methods to determine, e.g., when the paw is moving versus still? Figure 4A shows that the answer is yes if we use the DGP output: a two-state auto-regressive hidden Markov model (ARHMM; fit via Gibbs sampling on 1000 frames output from either DGP or DLC; [35]) performs well with no further pre- or post-processing. In contrast, the multiple DLC "glitches" visible in Figure 2 contaminate the segmentation based on the DLC traces, resulting in unreliable segmentation. See the video for further details. Similar results were obtained when fitting models with more than two states (data not shown).

**Conditional convolutional autoencoder (CAE) for more interpretable low-dimensional representation learning**. As a second downstream application, we consider unsupervised dimensionality reduction of behavioral videos [3, 1, 36, 37]. This approach, which typically uses linear methods like singular value decomposition (SVD), or nonlinear methods like convolutional autoencoders (CAEs), does not require user effort to label video frames. However, interpreting the latent features of these models can be difficult [38, 39], limiting the scientific insight gained by using these models. A hybrid approach that combines supervised (or semi-supervised) object tracking with unsupervised CAE training has the potential to ameliorate this problem [40, 41, 42, 43] – the tracked targets encode information about the location of specific body parts, while the estimated CAE latent vectors encode the remaining sources of variability in the frames. We refer to this ideal partitioning of variability into

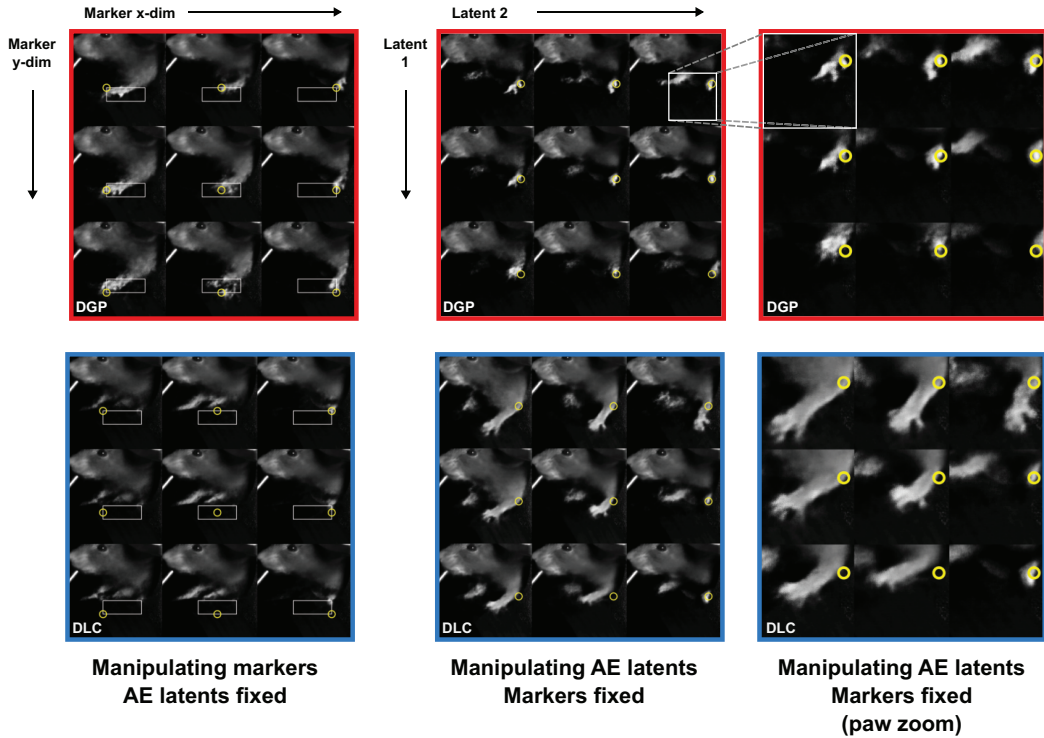

**Manipulating markers AE latents fixed** · **Manipulating AE latents Markers fixed** · **Manipulating AE latents Markers fixed (paw zoom)**

Figure 5: **Conditioning CAEs on DGP markers, but not DLC markers, leads to disentangled latents.** We incorporated the DLC and DGP markers into conditional CAEs trained on the mouse-wheel dataset. All frames are generated from 2-latent networks. **Left**: frames generated from the CAEs when changing the x and y coordinates of the left paw marker (yellow circle) for a given frame, with all other latents/markers fixed (white bounding box denotes the range of x/y coordinates). This manipulation should lead to noticeable changes in left paw position if markers are disentangled from latents. The network trained with DGP markers affords a much higher degree of control and produces more realistic looking images than that trained with DLC. **Center**: frames generated from the CAEs when changing the latents, with all markers fixed (white bounding box denotes the crop used for the right panels). This manipulation should not change the left paw position, but rather vary other (untracked) features of the image. Changes in the DGP reconstructions are limited to a small region around the tracked paws (yellow circle denotes left paw marker; see right panels for crop), demonstrating that the latents are encoding more local information such as paw configuration. DLC reconstructions show undesirable large movements of the left paw, demonstrating that the latents are encoding information about this tracked body part that should be present in the markers. **Right**: zoom of cropped region around the original paw location for frames in the center panel. See appendix Figure S1 for a more detailed quantitative analysis of latent/marker disentanglement.

more interpretable subspaces as "disentangling." Below we show that these hybrid models produce features that are more disentangled when trained with the output from DGP compared to DLC.

We fit conditional CAEs that take the markers output by DLC or DGP (hereafter referred to as CAE-DLC and CAE-DGP, respectively) as conditional inputs into both the encoding and decoding networks of the CAE, using the mouse-wheel dataset with 13 randomly chosen labeled frames (see Section S2 for implementation details). For this analysis, to obtain useful information across the full image, we labeled the left paw, right paw, tongue, and nose, rather than the four fingers on the left paw as in the previous section. Incorporating the tracking output from either method decreases the mean square error (MSE) of reconstructed test frames, for a given number of latents (Figure 4B). Furthermore, the networks trained with DGP outputs show improved performance over those trained with DLC outputs. Subsequent analyses are performed on the 2-latent networks, for easier visualization.

To test the degree of disentanglement between the CAE latents and the DGP or DLC output markers, we performed two different manipulations. First, we asked how changing individual markers affects the CAE reconstructions. We manipulate the x/y coordinates of a single marker while holding all other markers and all latents fixed. If the markers are disentangled from the latents we would expect to see the body part corresponding to the chosen marker move around the image, while all other features remain constant. We randomly chose a test frame and simultaneously varied the x/y marker values of the left paw (Figure 5, left). This manipulation results in realistic looking frames with clear paw movements in the CAE-DGP reconstructions, demonstrating that this marker information has been incorporated into the decoder. For the CAE-DLC reconstructions, however, this manipulation does not lead to clear movements of the left paw, indicating that the decoder has not learned to use these markers as effectively (a claim which is also supported by the higher MSE in the CAE-DLC networks, Figure 4B).

Second, we asked how changing the latents (rather than markers) affects the reconstructed frames. In this manipulation we simultaneously change the values of the two latents while holding all markers fixed. If the latents are disentangled from the markers we expect to see the tracked features remain constant while other untracked features change. For the CAE-DGP network this latent manipulation has very little effect on the tracked body parts, as desired (Figure 5, top center); instead, the manipulation leads to small changes in the configuration of the left paw (rather than its absolute location; Figure 5, top right). On the other hand, for the CAE-DLC network this latent manipulation has a large effect on the left paw location (Figure 5, bottom center), which should instead be encoded by the markers. These results qualitatively demonstrate that the CAE-DGP networks have better learned to disentangle the markers and the latents, a desirable property for more in-depth behavioral analysis. Furthermore, we find through an unbiased, quantitative assessment of disentangling, that using DGP markers in these models leads to higher levels of disentangling between latents and markers than DLC across many different animal poses present in this dataset (see Figure S1).

## 6 Discussion

In this work, we proposed a probabilistic graphical model built on top of deep neural networks, Deep Graph Pose (DGP), which leverages the rich spatial and temporal structures pervasive in behavioral videos. We also developed an efficient structured variational approach to perform inference in this model. The resulting semi-supervised model exploits information from both labeled and unlabeled frames to achieve significantly more accurate and robust tracking, using fewer labels. Our results illustrate how the smooth behavioral trajectories from DGP lead to improved downstream applications, including the discovery of behavioral "syllables," and interpretable or "disentangled" low-dimensional features from the behavioral videos.

An important direction for future work is to optimize the code to perform online inference for real-time experiments, as in [44]. We are currently integrating DGP on the "Neuroscience Cloud Analysis as a Service" (NeuroCAAS) platform [45], to help enable more scalable and reproducible analyses. Another important direction for future work is to extend our method to operate in 3D, fusing information from multiple cameras. Our variational inference approach should be extensible to this case, using similar epipolar constraints as in [25, 46] (using different inference approaches) to perform semi-supervised inference across views. In addition, [4, 5, 6] all use slightly different architectures and achieve similar accuracies. We plan to perform more experiments with the architectures from [5, 6] in the future. Finally, we would like to incorporate our model into existing toolboxes and GUIs to facilitate user access.

## Broader Impact

We propose a new method for animal behavioral tracking. As highlighted in the introduction and in [10], recent years have seen a rapid increase in the development of methods for animal pose estimation, which need to operate in a different regime than methods developed for human pose estimation. Our work significantly improves the state of the art for animal pose estimation, and thus advances behavioral analysis for animal research, an essential task for scientific discovery in fields ranging from neuroscience to ecology. Finally, our work represents a compelling fusion of deep learning methods with probabilistic graphical model approaches to statistical inference, and we hope to see more fruitful interactions between these rich topic areas in the future.

## Acknowledgments and Disclosure of Funding

We thank the authors of DeepLabCut [4] for generously sharing their code and data. This work was supported by grants from the Wellcome Trust (209558 and 216324) (LP), the Simons Foundation (LP, AN, NM, ES, JC, AW, MW), Gatsby Charitable Foundation GAT3708 (EB, AW, MW), the Searle Scholars Program (AB), Klingenstein-Simons Fellowship (AB), Sloan Foundation Fellowship (AB), Helen Hay Whitney Fellowship (ER), NIH grant NS116734 (AB), NIH Vision Sciences Training Grant EY013933 (CE), NIH T32 (MH015144) (ER), NIH U19NS104649 (Costa U19) (JC), NIH RF1MH120680 (Adesnik) (LP), NIH UF1NS107696 (Ji) (LP), NIH U19NS107613 (Miller U19) (LP, MW, EB, AW), NSF GRFP: DGE 16-44869 (NM), and NSF DBI-1707398 (Neuronex) (LP, JC, MW, EB, AW).

## Footnotes

[2] https://github.com/AlexEMG/DeepLabCut

[3]This exhaustive labeling was labor-intensive and we have not yet performed the same analysis for the other datasets in Table 1. As is visible in the appendix figures, our qualitative results are similar across all the datasets analyzed here; we plan to perform more exhaustive comparisons on other datasets in the future.

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
