[Supplementary Material]

# S1 Expanded methods

In this section we present our model and inference approach in fuller detail than was possible given space limitations in the main text. (To maintain the logical flow, in some cases we repeat points that were made in the main text methods.)

## S1.1 Deep Graph Pose model

The graphical model of DGP is summarized in Figure 1. We observe frames $x_t$ indexed by $t$, along with a small subset of labeled markers $y_{t,j}$ (where $j$ indexes the different targets we would like to track). The target locations $y_{t,j}$ on most frames are unlabeled, but we have several sources of information to constrain these latent variables: temporal smoothness constraints between the targets $y_{t,j}$ and $y_{t+1,j}$, which we capture with quadratic potentials $\phi_t$; spatial constraints between the targets $y_{t,i}$ and $y_{t,j}$, modeled with quadratic potentials $\phi_s$; and information from the image $x_t$, modeled by potentials $\phi_n$ parametrized by neural networks.

First let's define the potential function $\phi_n$ between the input image $x$ and the target's 2D location $y$. We define $f_\theta(\cdot)$ as a stack of a fixed pretrained ResNet-50 network and a trainable ConvNet parametrized by $\theta$. $f_\theta(\cdot)$ takes a frame $x$ as the input and outputs a 2D affinity map image, which ideally has a sharp peak at the most likely coordinates of the target. We then denote the sigmoid function as $\sigma(\cdot)$, and refer to $\sigma(f_\theta(x))$ as a "confidence map."

With the potential $\phi_n$, our target is to match this 2D confidence map to a 2D Gaussian bump centered at $y$ by minimizing the sigmoid cross entropy. Now let's define the Gaussian bump. We construct a bivariate Gaussian function with mean $y = [y_m, y_n]$ and variance $l^2$. The Gaussian function at the $m$th row and the $n$th column is

$$G(y, l^2)_{mn} = \frac{1}{2\pi l^2} \exp\left(-\frac{1}{2l^2}(m - y_m)^2 - \frac{1}{2l^2}(n - y_n)^2\right). \tag{S4}$$

The variance parameter was set as $l^2 = 1$ in practice.

The potential function $\phi_n$ for the $m$th row and the $n$th column entry in $\sigma(f_\theta(x))$ is defined as

$$
\begin{aligned}
\phi_n(y, x)_{mn} &= \frac{1}{2} w_n \left(-G(y, l^2)_{mn} \cdot \log(\sigma(f_\theta(x))_{mn}) - (1 - G(y, l^2)_{mn}) \cdot \log(1 - \sigma(f_\theta(x))_{mn})\right) \\
&= \frac{1}{2} w_n \left(-f_\theta(x)_{mn} \cdot G(y, l^2)_{mn} + f_\theta(x)_{mn} + \log(1 + \exp(-f_\theta(x)_{mn}))\right).
\end{aligned}
$$

Summing over all entries in the confidence map, we get the neural network potential $\phi_n$ as

$$\phi_n(y, x) = \sum_{m,n} \frac{1}{2} w_n \left(-f_\theta(x)_{mn} \cdot G(y, l^2)_{mn} + f_\theta(x)_{mn} + \log(1 + \exp(-f_\theta(x)_{mn}))\right).$$

We will write everything in vector form hereafter. We define $\mathbf{f}$ as the vectorized $f_\theta(x)$, define $\mathbf{h}$ as the vectorized $f_\theta(x) + \log(1 + \exp(-f_\theta(x)))$, and define $\mathbf{G}(y, l^2)$ as the vectorized $G(y, l^2)$, which is a function of mean $y$ and variance $l^2$. Thus, for each target $j$ we can rewrite the $j$-th image-based potential $\phi_n^j$ as

$$\phi_n^j(y_{t,j}, x_t) = \frac{1}{2} w_n(-\mathbf{f}_{t,j}^\top \mathbf{G}(y_{t,j}, l^2) + \mathbf{h}_{t,j}^\top \mathbf{1}), \tag{S5}$$

where $j$ is the index for target $j$ and $t$ is the index for frame $t$.

We use a simple quadratic potential $\phi_t$ to impose temporal smoothness:

$$\phi_t^j(y_{t,j}, y_{t+1,j}) = \frac{1}{2} w_t^j ||y_{t,j} - y_{t+1,j}||^2, \tag{S6}$$

which penalizes the distance between targets in consecutive frames; the weights $w_t^j$ in general may depend on the target index $j$, and can also vary in time. A more sophisticated version of the temporal clique could be an $L_2$ norm over the second or third order temporal difference, similar to optical flow.

The spatial potential $\phi_s$ is more dataset-dependent and can be chosen depending on the constraints that the markers should satisfy. Typical examples include a soft constraint that the paw marker should not exceed some distance from the elbow marker, or the nose should always stay within a certain radius of a static waterspout. This can be achieved by a soft-thresholding quadratic loss, leading to

a smooth pairwise potential between these two markers. Again, we start with a simple quadratic potential to encode these soft constraints:

$$\phi_s^{ij}(y_{t,i}, y_{t,j}) \;=\; \frac{1}{2} w_s^{ij} ||y_{t,i} - y_{t,j}||^2, \tag{S7}$$

which penalizes the distance between "connected" targets $y_{t,i}$ and $y_{t,j}$ (where the user can pre-specify pairs of connected targets that should have neighboring locations in the frame, e.g. paw and elbow).

We want to "let the data speak" and avoid oversmoothing, so the penalty weights $w_s$ and $w_t$ should be small. In practice we found that the temporal weights $w_t^j$ could be set using optical flow [27] which captures the vector field between neighbor frames. We first computed the vector field between two neighbor frames $t-1$ and $t$ using optical flow. Then we calculated the average motion vector for target $j$ from frame $t-1$ to frame $t$. The magnitude of the motion vector was denoted as $m_t^j$. Finally $w_t^j = \xi/m_t^j$, where $\xi$ is a constant scalar independent of dataset, time and target indices. The intuition is the larger the movement of the target is, the smaller the temporal clique weight should be. We set the spatial weights as $w_s^{ij} = c/d_{ij}$, where $d_{ij}$ is a rough estimate of the average distance (in pixels) between targets $i$ and $j$ and $c > 0$ is a small scalar (again independent of dataset and target indices $i, j$), led to robust results without any need to fit extra parameters.

We summarize the parameter vector as $\beta = \{\theta, w_n, w_t, w_s\}$, where $\theta$ denotes the neural net parameters. Given $\beta$ and the full collection of images $x$, the joint probability distribution over targets $y$ is

$$p(y|x, \beta) = \frac{1}{Z(x,\beta)} \exp\left( - \underbrace{\sum_{t=1}^{T} \sum_{j=1}^{J} \phi_n^j(y_{t,j}, x_t)}_{\text{neural network}} - \underbrace{\sum_{t=1}^{T-1} \sum_{j=1}^{J} \phi_t^j(y_{t,j}, y_{t+1,j}) - \sum_{t=1}^{T} \sum_{i,j \in \mathcal{E}} \phi_s^{ij}(y_{t,i}, y_{t,j})}_{\text{Gaussian graphical model}} \right),$$
$$\tag{S8}$$

where $\mathcal{E}$ denotes the edge set of constrained targets (i.e., the pairs $i, j$ with a nonzero potential function), $Z(x, \beta) = \int p(y|x, \beta) dy$ is the normalizing constant marginalizing out $y$, $T$ denotes the total number of frames, and $J$ denotes the total number of targets. The joint distribution can be described as a combination of a neural network component and a probabilistic graphical model over the latent variables (the unobserved targets $y$).

## S1.2 Structured variational inference

Our goal is to estimate $p(y^h \mid y^v, x, \beta)$, the posterior over locations of unlabeled targets $y^h$, given the frames from the video $x$, the locations of the labeled markers $y^v$, and the parameters $\beta$. (Here $h$ denotes hidden, for the unlabeled data, and $v$ denotes visible, for the labeled data.) Calculating this posterior distribution exactly is intractable, due to the highly nonlinear potentials $\phi_n$. We chose to use structured variational inference, similar to [29], to approximate this posterior. We approximate $p(y^h, y^v \mid x, \beta)$ with a Gaussian graphical model (GGM) with the same graphical model as Figure 1. We denote the approximate posterior as $q(y^h, y^v|x, \beta_q)$ ($\beta_q$ encodes variational parameters). To obtain a fully Gaussian variational approximation, we replace the neural network potentials $\phi_n^j$ with quadratic terms

$$\hat{\phi}_n^j(y_{t,j}, x_t) = \frac{1}{2} w_{n,q}^{t,j} ||y_{t,j} - \mu_n^{t,j}(x_t)||^2. \tag{S9}$$

Here the precision variables $w_{n,q}^{t,j}$ and means $\mu_n^{t,j}$ are variational parameters that we could optimize over independently. However, we found it more efficient to model the means $\mu_n$ as $\mu_n^{t,j}(x_t) = \sum_{m,n} \alpha_{mn} \text{Softmax}(f_\gamma^j(x_t))_{mn}$, where $\alpha_{mn} = [m, n]$. Here $f_\gamma(\cdot)$ is an inference neural network with parameters $\gamma$ whose output is a 2D affinity map, similar to $f_\theta(\cdot)$. Putting the pieces together, we have the fully Gaussian approximate posterior

$$q(y^h, y^v|x, \beta_q) = \frac{1}{\hat{Z}(x,\beta_q)} \exp\left( - \underbrace{\sum_{t=1}^{T} \sum_{j=1}^{J} \hat{\phi}_n^j(y_{t,j}, x_t)}_{\text{inference network}} - \underbrace{\sum_{t=1}^{T-1} \sum_{j=1}^{J} \phi_t^j(y_{t,j}, y_{t+1,j}) - \sum_{t=1}^{T} \sum_{i,j \in \mathcal{E}} \phi_s^{ij}(y_{t,i}, y_{t,j})}_{\text{identical to equation S8}} \right),$$
$$\tag{S10}$$

where $\hat{Z}(x, \beta_q)$ is the normalizing constant (which can be computed explicitly, due to the fully-Gaussian form of $q$), and $\beta_q = \{\gamma, w_{n,q}, w_t, w_s\}$.

Since $q(y^h, y^v | x, \beta_q)$ is a GGM, we can rewrite eq. S10 in the standard Gaussian form

$$
\begin{aligned}
q(y^h, y^v | x, \beta_q) &= \mathcal{N}(\mu_a, \Sigma_a) \\
\Sigma_a &= (\Sigma_s^{-1} + \Sigma_t^{-1} + \Sigma_n^{-1})^{-1} \quad (\text{S11}) \\
\mu_a &= \Sigma_a \Sigma_n^{-1} \mu_n \quad (\text{S12})
\end{aligned}
$$

where $\Sigma_n^{-1}$, $\Sigma_t^{-1}$ and $\Sigma_s^{-1}$ are the precision matrices corresponding to the potentials in $q(y^h, y^v | x, \beta_q)$; these have the form

$$
\Sigma_n^{-1} = \begin{bmatrix}
w_{n,q}^{1,1} & 0 & 0 & 0 & 0 & 0 \\
0 & w_{n,q}^{1,2} & 0 & 0 & 0 & 0 \\
0 & 0 & w_{n,q}^{2,1} & 0 & 0 & 0 \\
0 & 0 & 0 & w_{n,q}^{2,2} & 0 & 0 \\
0 & 0 & 0 & 0 & w_{n,q}^{3,1} & 0 \\
0 & 0 & 0 & 0 & 0 & w_{n,q}^{3,2}
\end{bmatrix} \quad (\text{S13})
$$

$$
\Sigma_t^{-1} = \begin{bmatrix}
w_t^1 & 0 & -w_t^1 & 0 & 0 & 0 \\
0 & w_t^2 & 0 & -w_t^2 & 0 & 0 \\
-w_t^1 & 0 & 2w_t^1 & 0 & -w_t^1 & 0 \\
0 & -w_t^2 & 0 & 2w_t^2 & 0 & -w_t^2 \\
0 & 0 & -w_t^1 & 0 & w_t^1 & 0 \\
0 & 0 & 0 & -w_t^2 & 0 & w_t^2
\end{bmatrix} \quad (\text{S14})
$$

$$
\Sigma_s^{-1} = \begin{bmatrix}
w_s & -w_s & 0 & 0 & 0 & 0 \\
-w_s & w_s & 0 & 0 & 0 & 0 \\
0 & 0 & w_s & -w_s & 0 & 0 \\
0 & 0 & -w_s & w_s & 0 & 0 \\
0 & 0 & 0 & 0 & w_s & -w_s \\
0 & 0 & 0 & 0 & -w_s & w_s
\end{bmatrix}. \quad (\text{S15})
$$

Thus the mean and covariance of the variational distribution $q(y^h, y^v | x, \beta_q)$ are $\mu_a$ and $\Sigma_a$, where $\mu_a$ is a function of $\gamma$, $w_{n,q}$, $w_t$, and $w_s$, and $\Sigma_a$ is a function of $w_{n,q}$, $w_t$, and $w_s$.

Let $P_h$ and $P_v$ denote the permutation matrices that map the vector $y$ to $y^h$ and $y^v$ respectively, i.e.,

$$
y^h = P_h y, \quad y^v = P_v y. \quad (\text{S16})
$$

Due to the Gaussianity of the joint distribution, we can write down the closed-form expression for $q(y^h | y^v, x, \beta_q)$ as

$$
q(y^h | y^v, x, \beta_q) = \mathcal{N}(\mu_h, \Sigma_h), \quad (\text{S17})
$$

where

$$
\begin{aligned}
\Sigma_h &= (P_h \Sigma_a^{-1} P_h^\top)^{-1}, \quad (\text{S18}) \\
\mu_h &= P_h \mu_a - (P_h \Sigma_a^{-1} P_h^\top)^{-1} P_h \Sigma_a^{-1} P_v^\top (y^v - P_v \mu_a). \quad (\text{S19})
\end{aligned}
$$

### S1.2.1 Evidence Lower Bound (ELBO)

Given the approximate posterior (eq. S17), and abbreviating $q(y^h) = q(y^h \mid y^v, x, \beta_q)$, we can now write down the evidence lower bound (ELBO) as

$$
\begin{aligned}
\mathcal{L} &= \mathbb{E}_{q(y^h)}[-\log q(y^h) + \log p(y^h, y^v | x, \beta)] \\
&= \mathbb{E}_{q(y^h)}[\log p(y^h, y^v | x, \beta)] + H(q) \\
&= -\mathbb{E}_{q(y^h)}\left[\sum_{t=1}^{T}\sum_{j=1}^{J}\phi_n^j(y_{t,j}, x_t)\right] - \mathbb{E}_{q(y^h)}\left[\sum_{t=1}^{T-1}\sum_{j=1}^{J}\phi_t^j(y_{t,j}, y_{t+1,j})\right] - \mathbb{E}_{q(y^h)}\left[\sum_{t=1}^{T}\sum_{i,j\in\mathcal{E}}\phi_s^{ij}(y_{t,i}, y_{t,j})\right] \\
&\quad - \log Z(x, \beta) + H(q) \\
&= \sum_{\substack{t=1, j=1 \\ t,j\in\mathcal{V}}}^{T,J}\frac{w_n}{2}(-\mathbf{f}_{t,j}^\top \mathbf{G}(y_t, l^2) + \mathbf{h}_{t,j}^\top \mathbf{1}) + \sum_{\substack{t=1, j=1 \\ t,j\in\mathcal{H}}}^{T,J}\frac{w_n}{2}(-\mathbf{f}_{t,j}^\top \mathbf{G}(\mu_{ht}, l^2 + \Sigma_{htt}) + \mathbf{h}_{t,j}^\top \mathbf{1}) \\
&\quad - \mathrm{Tr}((\Sigma_s^{-1} + \Sigma_t^{-1})P_h^\top \Sigma_h P_h) - \frac{1}{2}\mathrm{Tr}((P_h^\top \mu_h + P_v^\top y^v)^\top (\Sigma_s^{-1} + \Sigma_t^{-1})(P_h^\top \mu_h + P_v^\top y^v)) \\
&\quad - \log Z(x, \beta) + \log|\Sigma_a| - \log|P_v \Sigma_a P_v^\top|,
\end{aligned} \tag{S20}
$$

where $\mathcal{V}$ and $\mathcal{H}$ denote the sets of visible targets in visible frames and hidden targets in all frames respectively.

The bottlenecks of the ELBO computation in the full DGP model are $\Sigma_a \Sigma_n^{-1}\mu_n$, $\log|\Sigma_a|$, and $\mathrm{diag}(\Sigma_a)$, where $\Sigma_a \in \mathbb{R}^{TJ \times TJ}$ and $\Sigma_a^{-1}$ is a block tridiagonal matrix. All of these terms can be computed via message passing with $O(TJ^3)$ time complexity, due to the chain structure of the graphical model (and the corresponding block tridiagonal structure of the precision matrix). We used standard message passing algorithms to handle the required block tridiagonal matrix computations [47, 48, 49].

### S1.2.2 Semi-supervised DLC

To understand the various terms in the ELBO above it is helpful to start with a simpler special case. If we turn off the temporal and spatial potentials in eq. S20 (i.e., set $w_t = w_s = 0$) we arrive at the DGP-semi model discussed in the Results section. The corresponding ELBO is

$$
\begin{aligned}
\mathcal{L} &= \sum_{\substack{t=1, j=1 \\ t,j\in\mathcal{V}}}^{T,J}\frac{w_n}{2}(-\mathbf{f}_t^\top \mathbf{G}(y_t, l^2) + \mathbf{h}_t^\top \mathbf{1}) + \sum_{\substack{t=1, j=1 \\ t,j\in\mathcal{H}}}^{T,J}\frac{w_n}{2}(-\mathbf{f}_t^\top \mathbf{G}(\mu_{ht}, l^2 + \Sigma_{htt}) + \mathbf{h}_t^\top \mathbf{1}) \\
&\quad - \log Z(x, \beta) + \log|\Sigma_n| - \log|P_v \Sigma_n P_v^\top|,
\end{aligned} \tag{S21}
$$

where $\Sigma_h = (P_h \Sigma_n P_h^\top)^{-1}$ and $\mu_h = P_h \mu_n$. The first term is a conventional DLC-type cross entropy for labeled frames. The second term is a semi-supervised cross entropy for unlabeled frames. Instead of having the true marker locations for unobserved frames, we construct the Gaussian function using the 2D location output from the neural net. The second term encourages the confidence map $f_\theta$ to be unimodal to match the Gaussian approximate posterior. This semi-supervised term leads to better performance of DGP-semi compared to the original fully-supervised DLC (Figure 3).

### S1.3 Implementation details

There are a few issues regarding the optimization of the full DGP model that we considered during the implementation:

- In eq. S20, the log normalization term $\log Z(x, \beta)$ involves an integration over all frames and markers which makes the optimization intractable. In eq. S21, the graphical model factorizes over markers $j$ and frames $t$, which means that we can calculate the log normalization term $\log Z(x, \beta)$ directly by summing over pixels for each $t$ and $j$. But the summation over all pixels consumes a lot of time. In practice, we found that dropping the $\log Z$ term did not affect the results significantly.

- The optimization of the full ELBO involves two steps – expectation (E) and maximization (M). We estimate a good $q$ distribution in the E-step and optimize the network parameters given the $q$ distribution in the M-step. However, we found that in practice the E-step for the full DGP model was very time-consuming and only marginally improved the performance. Thus during the implementation, we simplified the $q$ distribution by dropping the temporal and spatial cliques in eq. S10, but still kept the full graph for $p$ as in eq. S8. Equivalently, this simplified the ELBO as well. The $q$ distribution optimized in the E-step can now factorize over frames thus making the computation a lot faster.

- With the simplified ELBO, the unknown parameters are $\{\theta, \gamma, w_n, w_t, w_s, w_{n,q}\}$. $\gamma$ and $w_{n,q}$ are the unknown parameters we need to learn during the E-step. We found that setting $\gamma = \theta$ led to good results and reduced the number of parameters. Moreover, optimizing $w_{n,q}$ didn't significantly improve the performance. Thus, for computational considerations, we decided to skip the E-step and set $w_{n,q} = 1$, which is a reasonable precision for the Gaussian bump, and set $\gamma = \theta$. In the M-step, we fixed $w_t^j$ using optical flow which provided the vector field of the dynamics between two neighbor frames, as described earlier. We set $w_s^{ij} = c/d_{ij}$, where $d_{ij}$ is the average distance (in pixels) between targets $i$ and $j$ and $c > 0$ is a small scalar (independent of dataset and target indices $i, j$); this led to robust results without any need to fit extra parameters. We also differentiated $w_n$ to be $w_n^v$ and $w_n^h$ for visible and hidden frames. Empirically, $w_n^h = 3$ and $w_n^v = 2w_n^h T/T_v$ ($T_v$ is the number of visible frames) led to good results; this upweighted the strength of labeled frames relative to unobserved frames. Therefore, the only parameter left is $\theta$.

- When simplifying the objective function, we can get rid of the harsh contraints on the form of the temporal and spatial cliques. The reason we choose both to be $L_2$ norms as in eq. S6 and S7 is that only $L_2$ norms in the $q$ distribution can lead to a closed-form expectation in the ELBO. However, if we don't consider these two cliques in the $q$ distribution, we can allow arbitrary forms. In the experiment, we still employed eq. S6 for the temporal clique, but employed a soft-thresholding quadratic loss $\phi_s^{ij}(y_{t,i}, y_{t,j}) = \frac{1}{2}w_s^{ij} Relu\left[||y_{t,i} - y_{t,j}||^2 - d_{ij}\right]$ for the spatial clique, where $d_{ij}$ is the average distance (in pixels) between targets $i$ and $j$. This spatial clique penalizes two markers when their distance is above the average distance calculated from the ground truth labels.

Therefore, the final objective function is

$$
\begin{aligned}
\mathcal{L}(\theta) \;=\; & \sum_{\substack{t=1,j=1 \\ t,j \in \mathcal{V}}}^{T,J} \frac{w_n}{2}(-\mathbf{f}_{t,j}^\top \mathbf{G}(y_t, l^2) + \mathbf{h}_{t,j}^\top \mathbf{1}) + \sum_{\substack{t=1,j=1 \\ t,j \in \mathcal{H}}}^{T,J} \frac{w_n}{2}(-\mathbf{f}_{t,j}^\top \mathbf{G}(\mu_{ht}, l^2 + \Sigma_{htt}) + \mathbf{h}_{t,j}^\top \mathbf{1}) \\
& -\frac{1}{2}\sum_{t=1}^{T-1}\sum_{j=1}^{J} w_t^j ||\mu_{t,j} - \mu_{t+1,j}||^2 - \sum_{t=1}^{T}\sum_{i,j \in \mathcal{E}} w_s^{ij} Relu\left[||\mu_{t,i} - \mu_{t,j}||^2 - d_{ij}\right], \quad \text{(S22)}
\end{aligned}
$$

where $\mu_{t,j} = y_{t,j}$ if $(t,j)$ is labeled; $\mu_{t,j} = \mu_{ht,j}$ otherwise. We maximized the above objective function and calculated the gradients for $\theta$ using standard automatic differentiation tools, and performed standard stochastic gradient updates to estimate these parameters.

## S2    Conditional convolutional autoencoder

### S2.1    Implementation details

We fit conditional convolutional autoencoders (conditional CAEs) on 192x192 grayscale images from [30]. In addition, we used 4 markers output by DLC/DGP: left paw, right paw, tongue, and nose. To condition the encoder network on these values we turned each marker into a one-hot 2D array and concatenated these with the corresponding frame, so that the input to the encoder was of size $(192, 192, 5)$. To condition the decoder network on these values we first centered the marker values by subtracting their median (computed over the entire dataset) and then concatenated these values to the latents before feeding them into the decoder. See Table S1 for network architecture details. We trained the autoencoders by minimizing the MSE between original and reconstructed frames using the Adam optimizer [50] with a learning rate of $10^{-4}$, a batch size of 100, and no regularization. Models were trained for 300 epochs.

| Layer | Type | Channels | Kernel Size | Stride Size | Zero Padding | Output Size |
|---|---|---|---|---|---|---|
| 0 | conv | 32 | (5, 5) | (2, 2) | (1, 2, 1, 2) | (96, 96, 32) |
| 1 | conv | 64 | (5, 5) | (2, 2) | (1, 2, 1, 2) | (48, 48, 64) |
| 2 | conv | 128 | (5, 5) | (2, 2) | (1, 2, 1, 2) | (24, 24, 128) |
| 3 | conv | 256 | (5, 5) | (2, 2) | (1, 2, 1, 2) | (12, 12, 256) |
| 4 | conv | 512 | (5, 5) | (2, 2) | (1, 2, 1, 2) | (6, 6, 512) |
| 5 | dense | $N$ | NA | NA | NA | $(1, 1, N)$ |
| 5 | concatenate | NA | NA | NA | NA | $(1, 1, N+2M)$ |
| 6 | dense | 36 | NA | NA | NA | (1, 1, 36) |
| 7 | reshape | NA | NA | NA | NA | (6, 6, 1) |
| 8 | conv transpose | 256 | (5, 5) | (2, 2) | (1, 2, 1, 2) | (12, 12, 256) |
| 9 | conv transpose | 128 | (5, 5) | (2, 2) | (1, 2, 1, 2) | (24, 24, 128) |
| 10 | conv transpose | 64 | (5, 5) | (2, 2) | (1, 2, 1, 2) | (48, 48, 64) |
| 11 | conv transpose | 32 | (5, 5) | (2, 2) | (1, 2, 1, 2) | (96, 96, 32) |
| 12 | conv transpose | 1 | (5, 5) | (2, 2) | (1, 2, 1, 2) | (192, 192, 1) |

Table S1: Conditional CAE architecture for the mouse-wheel dataset using $N$ latents and $M$ markers (each with an x and y value, for a total of *2M* marker dimensions). Kernel size and stride size are defined as (x pixels, y pixels); padding size is defined as (left, right, top, bottom); output size is defined as (x pixels, y pixels, channels).

## S2.2 Disentangling analysis

The disentangling analyses presented in Figure 5 require fixing some inputs to the network while varying others. Below we describe this manipulation in more detail. We performed these manipulations on 2-latent networks to make visualization in the latent space easier.

*Manipulating markers.* We chose a random test frame and varied the x/y coordinates for a specific marker (left paw). The limits of the x/y values were the $10^{th}$ (minimum) and $90^{th}$ (maximum) percentiles of the DGP outputs on the test set for the specified marker. We did not allow different limits for the DLC/DGP networks, in order to make the comparison more direct. After choosing x/y values for the specified marker we converted these into a one-hot 2D array, as with the other (unchanged) markers from the chosen frame. These one-hot 2D arrays were concatenated with the original frame and then fed into the CAE encoder to produce the latents. The latents were then concatenated with the median-subtracted marker values (one of which is being changed, the rest of which stay the same). This vector was then pushed through the decoder network to produce the reconstructions.

Note that in this conditional architecture the latents themselves are marker-dependent, so are not truly held fixed. We also fit conditional CAE architectures where just the decoder was conditioned on the markers, and the encoder only used the frame as input. We found the results from the disentangling analysis to be qualitatively similar, though reconstructions generally looked cleaner with the architectures that incorporated conditioning in both the encoder and decoder networks (data not shown).

*Manipulating latents.* We chose a random test frame and this time varied the latents while keeping the marker values fixed. Similar to above, we used the $10^{th}$ (minimum) and $90^{th}$ (maximum) percentiles of the latents on the test set as limits (this time allowing different limits for each DLC/DGP architecture). We then concatenated the new latent values with the marker values from the original frame, and pushed this vector through the decoder network to produce the reconstructions.

*Quantifying disentanglement.* To quantify the disentanglement results from Figure 5 (center, right panels), we chose the left paw as a (tracked) target of interest that should ideally not undergo large changes when manipulating the latents. If disentanglement is high (which we desire), the differences between the generated paw and the original paw should be small. For each image generated from the latent manipulation, we take a small crop around the original location of the left paw and compute the MSE between this generated paw and the original paw.

Figure S1: **Conditional CAEs display better disentangling properties when using DGP rather than DLC markers.** **A**: A total of 64 representative frames were chosen from the test set by performing k-means on the 2-latent unconditional AE latents; pictured are 16 of those frames, which include the left paw (the target feature of this analysis) in multiple positions. **B**: For each frame, the markers are held fixed while the 2 latents are varied, producing a matrix of generated images. For each generated image we crop around the left paw. The paw position should remain stable throughout the latent manipulation if there is good disentanglement between latents and markers. Latent manipulations of two example frames are shown (in columns) for networks trained using DGP or DLC markers (in rows). Yellow circles indicate the left paw marker location. **C**: We compute MSE between each generated frame and the original frame around the paw, and average over all latent values. Small MSE indicates desired stability in the tracked paw. For almost all frames the DGP-trained network exhibits lower MSE, thus demonstrating a higher degree of disentangling.

We repeat this process for an unbiased sample of test frames. To obtain these frames we performed k-means clustering on the unconditional CAE latents. From each of 64 clusters we take the frame that is closest to the cluster centroid (Figure S1A), and perform the process of generating frames by evenly sampling the latent space - 4 grid points along each of 2 dimensions, for a total of 16 generated frames. We compute the MSE in crops around the labeled paw position as described above (Figure S1B, C). We find that on average the CAE-DGP networks have lower MSE, indicating that these disentanglement results generalize to many other paw positions (and therefore marker values) found in this dataset.

Figure S2: **Different active learning strategies can speed up training of DGP without the need to query for new human labels.** Evolution of the RMSE during training using four different scoring strategies to query for pseudo targets (unlabeled frames) on the twomice-top-down dataset. The RMSE decays faster by employing active learning strategies to select unlabeled frames to train DGP.

## S3  Active learning

Given a video with a set $N$ of frames, let $M$ be the set of user-labeled frames, and let $U = N - M$ be the set of frames not selected to be labeled. Many previous animal pose estimation algorithms select $M \in N$ by applying $k$-means clustering to a randomly sampled subset of frames in the video, and then uniformly sampling frames from these clusters. The number of clusters for $k$-means is manually set usually as 10 or 25. Although this approach can work relatively well, several failure modes can occur. For example, if the animals are in fixed positions or not moving for extended periods of time (a common scenario in biological experimental settings), uniformly sampling frames from such a video will result in selecting many identical or redundant frames. Furthermore, if the frames of interest in the video—parts of the video where animals perform behaviors of interest to experimenters—are sparse, randomly sampling frames from a video is highly likely to overlook these frames. Additionally, even if the frames of interest are included during clustering, since these frames are scarce, they may be considered as outliers by a naive $k$-means algorithm and there are no guarantees that these frames of interest would be sampled from any clusters and selected for downstream labeling or training steps.

This problematic lack of diversity in the training set can be addressed by fully supervised algorithms by querying new labeled frames, i.e. by asking the user to manually label new frames based on the performance of the network after training. This process can be repeated several times until the network outputs are satisfactory. Previous work has shown that active learning by querying informative and representative samples can be more effective than passive human labeling [51, 52]. Most of this work proposes strategies to score and sample unlabeled frames for manual labeling.

Here we propose to employ these strategies to select not only the set of frames to be manually labeled by an experimenter or oracle $M$, but also to select the subset of unlabeled frames used during training, $S \in U$, which provide additional information to semi-supervised algorithms such as DGP. Unlike semi-supervised learning, which exploits what the learner thinks it knows about the data by employing, for example, unlabeled frames with pseudo targets during training, active learning exploits what the learner does not know about the data, by exploring the space or querying for information about the unlabeled frames [53]. These two approaches can be combined naturally; see [53] for a review.

| Scoring | Description |
|---|---|
| random | Frames are sampled at random from video |
| motion | Frames are sampled based on their motion energy; frames that are very similar to their nearest neighbors are sampled less |
| $H_{win}$ | Frames are sampled based on the difference between the entropy of mean predictions and the mean entropy of predictions [54] |
| $p_{max}$ | Frames are sampled based on their uncertainty; frames with more uncertain network outputs are re-sampled more frequently |

Table S2: Scoring strategies for active learning

We train DGP using the four different active learning strategies described in Table S2 to query for pseudo targets (unlabeled frames) employed during training. Figure S2 illustrates the evolution of the RMSE for a subset of hidden frames in the training set. We see that with active learning the loss converges faster than by using random sampling. In future work we plan to investigate these different active learning approaches in more detail and across additional datasets.

## S4    Results on all datasets

Figures S3-S6 show the comparisons between DLC and DGP on the mouse-reach, fly-run, twomice-top-down, and fish-swim datasets (Table 1). In each case, results were similar to those seen in Figure 2. Please check these videos to see the trace comparisons between DLC and DGP for these datasets.

We also did some additional experiments with the intermediate models between DLC and DGP, including DLC-ours, DGP-semi and DGP-spatial described in Table S3.

| Model | Description |
|---|---|
| DLC | original DLC implementation, binary target maps for cross entropy loss |
| DLC-ours | our DLC implementation, gaussian target maps for cross entropy loss |
| DGP-semi | DGP model without cliques, semi-supervised loss only |
| DGP-spatial | DGP model with only spatial clique |
| DGP | full DGP model with both spatial and temporal cliques |

Table S3: All the models we ran for comparison.

Table S4 summarizes the experimental setup for each model and each dataset. Note that the total number of iterations we ran DGP models was way less than the total number of iterations we ran using DLC. For example, we initialized DLC-ours from DLC after 200k; then initialized DGP using DLC-ours after 6k. Equivalently, we can assume that we initialized DGP using DLC after 206k iterations and ran another 80k (8k x 10) iterations with batch size 1. Thus, we ran about 280k iterations in total for DGP while we ran 1m (1,000,000) iterations for DLC.

Results are consistent across datasets: DGP and DGP-spatial achieve similar performance across all five datasets, but DGP has smoother traces; each tends to outperform DGP-semi, which in turn outperforms either implementation of DLC. We also provide full videos for the comparisons of 5 traces.

| Dataset | Model | Initialization | Number of iterations | Batch size |
|---|---|---|---|---|
| mouse-wheel | DLC | pre-trained ResNet | 1m | 1 |
| | DLC-ours | DLC 200k | 86k | 1 |
| | DGP-semi | DLC-ours 6k | 8k | 10 |
| | DGP-spatial | DLC-ours 6k | 8k | 10 |
| | DGP | DLC-ours 6k | 8k | 10 |
| mouse-reach | DLC | pre-trained ResNet | 1m | 1 |
| | DLC-ours | DLC 200k | 31k | 1 |
| | DGP-semi | DLC-ours 5k | 5k | 5 |
| | DGP-spatial | DLC-ours 5k | 5k | 5 |
| | DGP | DLC-ours 5k | 5k | 5 |
| fly-run | DLC | pre-trained ResNet | 1m | 1 |
| | DLC-ours | DLC 200k | 185k | 1 |
| | DGP-semi | DLC-ours 5k | 18k | 10 |
| | DGP-spatial | DLC-ours 5k | 18k | 10 |
| | DGP | DLC-ours 5k | 18k | 10 |
| twomice-top-down | DLC | pre-trained ResNet | 1m | 1 |
| | DLC-ours | DLC 200k | 114k | 1 |
| | DGP-semi | DLC-ours 2k | 12k | 10 |
| | DGP-spatial | DLC-ours 2k | 12k | 10 |
| | DGP | DLC-ours 2k | 12k | 10 |
| fish-swim | DLC | pre-trained ResNet | 1m | 1 |
| | DLC-ours | DLC 200k | 170k | 1 |
| | DGP-semi | DLC-ours 2k | 17k | 10 |
| | DGP-spatial | DLC-ours 2k | 17k | 10 |
| | DGP | DLC-ours 2k | 17k | 10 |

Table S4: Experimental setup for each dataset and each model. For DLC, we initialized it with a pre-trained ResNet from ImageNet and ran 1m iterations using 1 batch stochastic gradient descent (sgd) for all datasets. For the mouse-wheel dataset, we initialized DLC-ours from DLC after 200k iterations and ran it for 86k iterations with batch size 1. We initialized DGP-semi, DGP-spatial, and DGP from DLC-ours after 6k iterations and ran each for another 8k iterations with batch size 10. Likewise, we ran the other four datasets correspondingly.

Figure S3: Comparing DeepLabCut (DLC) versus Deep Graph Pose (DGP) on the mouse reaching dataset from [4]. Conventions and conclusions as in Figure 2.

Figure S4: Comparing DeepLabCut (DLC) versus Deep Graph Pose (DGP) on the fly ball-turning dataset from [32]. Conventions and conclusions as in Figure 2.

Figure S5: Comparing DeepLabCut (DLC) versus Deep Graph Pose (DGP) on the twomice-top-view dataset with two mice. Conventions and conclusions as in Figure 2.

Figure S6: Comparing DeepLabCut (DLC) versus Deep Graph Pose (DGP) on the swimming fish dataset from [33]. Conventions and conclusions as in Figure 2.