[Reviews · NeurIPS 2020]

Review 1

Summary and Contributions: In this work, the authors describe a system for the tracking of animal pose, which uses variational inference over probabilistic graphical models to achieve good results in a sparse labeling regime, where lots of data is available, but only some of it is labeled. The proposed method achieves results that are competitive with current state of the art approaches, even with a fraction of the labeled data.

Strengths: The application of animal pose estimation is an important one in systems and behavioral neuroscience, and there has been a minor renaissance in that field in recent years brought about through the advent of tools like Deep Lab Cut. The requirement to annotate large amounts of data in order to use these tools (even when leveraging transfer from other pre-trained models) is the achilles heel of these methods. Interestingly, the proposed method outperforms Deep Lab Cut even in the case of having most of the data available, which makes this method easy to choose. The perform of the new method is competitive with Deep Lab Cut with roughly 1/3 to 1/2 of the amount of data, which is a real win. In addition, the method automaticlaly discovers interpretable low dimensional representations of the animal's movement (e.g. resting states, different actions) which is itself a potentially valuable analysis tool for the sorts of analyses that animal pose estimation is used for. These kind of analyses aren't themselves new (and the authors cite relevant work, e.g. citation 26), though it is nice to bring those kinds of advantages directly to the estimation of pose.

Weaknesses: The paper makes excellent use of reasonably sophisticated modeling techniques, but it is a bit less clear what is novel from a methodological standpoint. Certainly, the framing of the paper focuses on the application, and it does it well, but I do wonder about the suitability for the NeurIPS audience as a result. To the extent that there are novel methodological contributions, those are not clear and not put in focus or discussed in the context on the literature. The paper does do a good job of comparing against an obvious competitor method (Deep Lab Cut), but again, the application is the focus. Thus I wonder if this wouldn't be more appropriate for a more application (or perhaps, neuroscience)-focused venue. I think at the end of the day, this is a decision for the area chair, though I'm personally a bit unsure, given how the paper is written..

Correctness: The claims and methodology appear to be sound.

Clarity: The paper is clearly written.

Relation to Prior Work: See weaknesses discussion above; most of the discussion of prior work focuses quite narrowly on the application, rather than methodological advances. This is fine for an application paper.

Reproducibility: Yes

Additional Feedback: Update post-rebuttal: I find the authors' explanation of their methodological contributions helpful, but it does not change the fact that this paper is effectively written for a different, more biology-focused audience, and thus those technical details get lost. I would even say that the paper is beautifully executed for that audience; however, I still find it an odd fit for the NeurIPS community, and believe it would be better suited to a different venue.


Review 2

Summary and Contributions: This paper presents a method to estimate 2d pose of animals, particularly focusing on the case that the annotations for training are scarce (which would be common in animal cases). For that, a graphical model built on top of neural network models is proposed to exploit temporal and spatial cues as a way to propagate known annotations to non-annotated frames and joints. Experiments show that the method shows better performance than previous work (DLC) that processes each frame separately.

Strengths: * It’s an interesting research problem with a convincing motivation, assuming 2D keypoint annotations for animals often are not sufficiently obtained (compared to humans). It looks like a reasonable and convincing decision to leverage the spatio-temporal cues to tackle the challenge. * The experiments are properly performed to support the strength of the paper. The ablation study Fig 3 is quite helpful to see the advantage of the proposed model. The experiments on downstream tasks are also convincing.

Weaknesses: - Spatio temporal models are quite common in human motion field, and the formations in Eq (1) and (2) look quite naive and less novel. In particular, these terms basically enforce that the joints in adjacent frames or the neighboring joints in a single frame to be the same point, which is not correct in general (e.g., these are not zeros even for GT data). These terms often conflict to the data measurement term (phi_n), making it difficult to balance between data term and prior term. I believe section 3 would be the core part of this paper but unfortunately it does not provide enough details. The paper also does not provide enough details for their training and testing data and its split. For example, in Fig 3, it seems like the same videos are used for training and testing, given a few sparse annotations (in the same video). I got confused that there exists any training stage given separate training sets. Also details for training/test procedures are missing. Do you pre-train the neural network somehow? I understood that additional inference is needed to be performed in testing time, but this was also unclear for me.

Correctness: Seems correct

Clarity: More details about the method, dataset, training/testing procedures should be provided.

Relation to Prior Work: The paper doesn’t cover enough related work. Even if the major domain is on animals, very similar techniques and ideas (e.g., spatio-temporal) have been used in human pose estimation domain. I believe it is meaningful to cover them. For example, Spatio-temporal Matching for Human Pose Estimation, Zhou 2016 Kronecker-Markov Prior for Dynamic 3D Reconstruction, Simon 2015

Reproducibility: No

Additional Feedback:


Review 3

Summary and Contributions: This paper provides a semi-superivsed way for animal 2D pose estimation from limited number of labels. In this domain, unlike 2D human pose estimation, the number of labels obtained are often limited. This paper proposes a graphical model over 3 consecutive frames that encourages temporal smoothness, structural consistency, and a keypoint detector network. The graphical model is approximated with a Gaussian graphical model, where each term has a Gaussian assumption. The temporal and structural terms already satisfy this, but this intoduces the heatmap keypoint prediction to be regularized to output a 2D guassian heatmap, which the paper shows as an effective regularization. The graphical model is optimized with variational inference. The paper shows that the proposed graphical model outperforms previous supervised approach, especially under low-data setting. The paper then verifies that the resulting improvement and robustness is in fact significant enough for downstream tasks. This is demonstrated through two experiments, one for temporal segemntation where DGP output provides more coherent segmentation, and another the predicted keypoint conditioned representation learning via CAE, where using the DGP keypoints leads to more disentangled representation, which shows it's reliability.

Strengths: - Proposes a simple but effective approach to make pose estimation more robust under low-labeled data regime. - The biggest strength of this paper is that it verifies that the obtained numerical improvemnt is in fact significant enough for later use. This verification exemplifies that the proposed approach is useful for downstream tasks, which is very imporant. I appreciate that the authors are considering the actual task, instead of only improving upon SOTA approach numerically. More papers should do this. - It is nice that at test time DGP can just use the neural network pose predictor without the temporal / structural priors without reducing accuracy. - Experiments are conducted over multiple runs with error bars. - Great problem that is not as studied and can have high impact in other disciplines who would like to use such tools for analysis.

Weaknesses: - How much does the spatial and temporal potentials matter? The paper conducts experiments on DLC semi (supervised + gaussian regularization) and DGP, however the influence of spatial and temporal potentials are not evaluated independently. This seems like an informative ablation study to do, especially since the paper claims the difference with prior work is that prior work does not consider temporal and spatial priors. There is a recent work OptiFlex by Liu et al which also uses temporal information, this should be cited. (Although the proposed approach is much simpler than OptiFlex) - The experimental protocol is good, but it is not compared against a previously published semi-supervised pose prediction approach. Only copared against a fully supervised method (DLC) and a baseline semi-supervised method which is an ablative version of the proposed approach (no temporal and structural priors). In the supplemental authors claim that this is a form of previous weakly-supervised approach, however the cited paper is from 1972 (is this a typo?). You would think there's another semi-supervised approach for animal 2D pose estimation, as in this area limited labeled data seems like a very common problem. However I am not an expert in animal 2D pose estimation so I do not have any papers to suggest. From what I know fo 2D human pose estimation this is not as studied as it is ok to assume there are lots labeled data. So possibly there is no other baseline. As I am not suggesting a paper to compare against, I am not taking points away from this, but if the authors can find a previously published work on this the paper may be more convincing. If this is the first instantiation the writing may also reflect this. - Writing is well written, however, much of the detail is in the supplementary that just from reading the paper the detail of the model eludes the reader. I think there are ways to incorporate some more information from the supplemental to the main paper. For example conditional CAE, it's easy to say that one just appends the detected keypoints as an image as additional channel. The detail of the gaussian prior on the heatmap detection also may be incorporated a little more.

Correctness: Yes. The verification of the proposed approach on downstream tasks is particularly nice.

Clarity: - Yes, in general it's very well written. - However one thing is unclear, what is alpha in the supplemental line 515, where does it come from?? - Also sentence in line 67, "We use the notation \phi_n to indicate that this potential is parametrized by a NN" this was confusing, is the "n" teh notation, or "\phi"? turns out it's both, can be simplified to: "We use NN to parametrize \phi_n".

Relation to Prior Work: Yes

Reproducibility: Yes

Additional Feedback: I think this is a solid paper that makes contribution in animal pose estimation. The proposed approach is simple and most importantly its effectiveness is evaluated in a downstream task where there seems to be tangible differences due to the blips. I think this experiment makes this paper a very nice work that others can build upon both experimentally (test on downstream task in addition to quantitative evaluation) and methodically (more robust APE under low-data regime). Note that the video and other blue links in the paper did not work for me. However, I am not an expert in animal 2D pose estimation, so I may not know of other papers that explore semi-supervised setting in order to deal with limited amount of labeled data. However I know that DLC is a very popular toolbox for this. I want to hear from other reviewers for any references I am missing, if not I am happy to bump up the rating. ---- After reading the author response + other reviews, I stand by my initial rating. Here are some response to other reviewers comments: - On the point of compare with PoseTrack: I find that problems dealing with animals have a very different set of challenges than that of humans. In particular in PoseTrack there are many human-to-human occlusions and cover quite a different aspect than the data biologists need to work with (low-resolution, imperfect lighting, and most importantly lack of large scale data). Therefore asking a method that are looking into challenges specific to animals to present a method that works equally well on PoseTrack seems to me a high ask and outside the scope of this paper. - I agree that writing is rather out of standard (esp from the lens of human pose estimation literature), however I believe it's the way papers are written in more bio focused background and while I agree that more should be in the main paper (as stated in the original paper) this is something authors can figure out in the camera ready. - Novelty on Method: I think the proposed method is a simple sensible model that is shown to work well. I am with R2 on this point, applying existing method to a new domain is equally important and it's nice to see that such a simple method does work and analyzed in depth. I think the analysis that they present in this paper is particularly nice, and hope all animal behavior papers will follow suit.


Review 4

Summary and Contributions: In this paper, the authors propose a probabilistic graphical model built on top of deep neural networks for animal pose tracking, which uses both spatial and temporal constraints and a new structured variational inference algorithm. In the experiments, the proposed approach obtains more accurate and robust tracking while using fewer labels.

Strengths: 1) This paper studies an interesting task of animal pose tracking.

Weaknesses: 1) The proposed method is not end-to-end. The authors are encouraged to try recent techniques such as graph neural networks for fusion of deep learning with graphical model. 2) The proposed framework is quite simple and the method novelty seems very limited. 3) The experiment settings are a little unclear and there are no comparisons with other animal pose estimation methods.

Correctness: The method seems correct but seems outdated.

Clarity: Overall this paper is easy to follow, but the paper is not well organized, the links in both the main text and supplementary material are invalid, the experimental setting and details should be included in the main text.

Relation to Prior Work: The authors did not discuss the literature of graph neural networks in the related work, but it should be put in the main text not the supplementary material.

Reproducibility: No

Additional Feedback: After reading the rebuttal and other reviewers' comments, I still have some major concerns for this paper 1) paper organization, 2) method technical soundness and 3) experiments. The original paper organization is not well structured and many key technical details of the proposed method are put in the supplementary material. As this paper stresses a semi-supervised deep graphical model, I don't think it should be organized in a neuroscience-application way. Also It is not clear how well the proposed framework could perform on human pose tracking and PoseTrack seems to be a popular benchmark in evaluation. I feel this task is probably easier than human pose tracking based on the visual results shown in the paper. This raises my concern that the proposed method could not compete with the popular methods (OpenPose, AlphaPose, HRNet, PoseFlow) and tweaking popular methods in human pose tracking could potentially perform fairly well in animal pose tracking. I understand the proposed method focuses on semi-supervised learning but the authors should at least provide more validations on PoseTrack to further convince the reader about the method effectiveness. I changed to my rating to 4 considering the efforts made in this paper but in my opinion this paper still needs fair amount of revision and polishment to reach the standard for a NeurIPS paper.

[Author Response · NeurIPS 2020]

We thank the reviewers for their time and effort reviewing our paper, and for the positive appraisal of its content. Our work is the first semi-supervised model to achieve SOTA results in a sparsely-labeled data regime on a wide variety of animal datasets. We firmly believe this work will make a valuable contribution to the NeurIPS community.

To begin, we emphasize a few critical points shared among reviewers:

1) **Methodological novelty.** Our model is a novel semi-supervised learning approach with spatial and temporal cliques for large-scale sparsely-labeled videos. In the field of human pose estimation (HPE), there have been significantly fewer methods in the video domain due to a limited number of large-scale benchmarks for video tracking. Thus the significance of semi-supervised models utilizing unlabeled frames is nonnegligible. Two papers in HPE closely related to our work are Song et al (CVPR2017) and Bertasius et al (NeurIPS2019). The former has the spatial-temporal graph but doesn't consider unlabeled frames, and the latter works for sparsely-labeled data but without any spatial constraint. Moreover, in the field of animal pose estimation (APE), we believe there is no such work yet. Another contribution is formulating the semi-supervised learning in a Bayesian framework. We employed a structured variational inference method for the hybrid Bayesian latent variable model which hasn't been done in this field. We apologize if these points were not clear in the main paper, and we will emphasize them more clearly in the revision.

2) **Lack of comparison to other approaches.** We would like to clarify that we use DLC as the underlying architecture for DGP to show that when such a model is recast in the proposed semi-supervised framework–which considers unlabeled frames and spatial and temporal cliques–the overall model performance improves. We didn't find other semi-supervised approaches in the APE literature that incorporate unlabeled frames, so instead our experiments consisted of ablation studies comparing different components of DGP. Our framework is easily compatible with other architectures; we have already run comparisons of DGP with a DPK (Graving et al elife2019) architecture instead of DLC, and will include these results in the revision.

3) **Missing details.** We apologize for missing some details. We aimed at writing a neuroscience-application focused paper, thus unavoidably moved some modeling and experiment details to the supplementary. We will definitely consider moving them back if it helps with model clarity and polish the experimental details in the revision.

4) **Missing link.** We would like to sincerely apologize for the missing link issue. The submission system wiped off the links and we didn't carefully check them after submission. We will fix this issue in the revised version.

**Reviewer 2:**

1) **Spatio-temporal models.** The focus of this work is not in the complexity of the spatio-temporal cliques, but rather in how to incorporate these cliques into a hybrid graphical model with latent variables. As mentioned in the paper, the proposed spatio-temporal cliques can include additional dynamics or other structural constraints; this is an important topic for future work. Here we provide a set of cliques which achieve good performance in a wide variety of datasets.

2) **Missing details in sec. 3 and experiments.** We will include more details for the training and testing splits in Fig 3. We split the data in two separate train and test sets. To train our model, we employed a pre-trained Resnet-50 on Imagenet. At test time, in DGP-NN we pass test frames into the trained network and calculate the 2D locations. During test time, we can optionally run one additional E-step optimization for DGP which functions as a post-hoc smoothing without changing the network parameters. We will clarify these points in the revised version.

3) **Related work.** We mainly respond to this point in **Methodological novelty**. The related papers pointed out by the reviewer didn't consider unlabeled frames. This discrepancy is non-trivial from a modeling perspective. However, we thank the reviewer for pointing out these references and will include them in the revised version.

**Reviewer 3:**

1) **How much do potentials matter.** The comparison between DLC semi and DGP indicates the benefit of the spatial and temporal cliques. But as the reviewer pointed out, we didn't show cross-ablation studies to highlight the impact of the individual cliques. We have already done these analyses and will include them in the revised version. We also thank the reviewer for pointing out the work by Liu et al and we will cite this paper in the revised version. In DGP, the proposed temporal clique can be extended to a more elaborate function e.g. optical flow. As mentioned earlier, our paper's focus is to propose a hybrid latent variable model where information from scarce human-labeled frames can be propagated to unlabeled frames. Our current construction of the spatial and temporal cliques is simple and flexible, but these can be extended to other more complex forms.

2) **Notation issues in the supplement.** We apologize for these issues. These will be corrected in the revised version.

**Reviewer 4:**

1) **The model is not end-to-end.** This seems to be a misunderstanding: our model is an end-to-end model. DGP combines a graphical model and a neural network in a hybrid model, where the loss function includes the parameters from both of these components. This is different from - and empirically superior to - training a baseline model (such as DLC) and applying a Kalman filter to the network outputs afterwards.

2) **Graph neural network.** The graph neural networks (GNN) literature that we have encountered pertains to networks where the different nodes have graph structures. In constrast, DGP attaches a CRF-like graph to the output of the neural network, as opposed to changing the network structure. We would ask the reviewer for additional clarification if we missed some GNN literature more closely related to our proposed model or if we misunderstood the point being made.

[Meta-Review · NeurIPS 2020]

This submission proposes a method animal 2D pose estimation and tracking given limited amounts of ground truth annotations. It initially received four reviews with diverging scores (5,6,7,4), which remained unchanged after the rebuttal. The reviewers appreciated importance of the application, solid empirical performance compared to DeepLabCut (including tests on downstream tasks) and insightful analysis of the learned representations. At the same time, the main concerns of the reviewers were limited methodological novelty beyond applying known methods to the new domain of animal tracking, as well as limitations in the empirical studies. This case was further discussed between the AC and the SAC, who arrived to the conclusion that the merits of this submission in advancing animal tracking outweigh its limitations. The final recommendation is to accept as a poster.